# Manufacture of SiC: Effect of Carbon Precursor

**DOI:** 10.3390/ma16052034

**Published:** 2023-03-01

**Authors:** Enrique V. Ramos-Fernández, Javier Narciso

**Affiliations:** 1Laboratorio de Materiales Avanzados, Departamento de Química Inorgánica, Instituto Universitario de Materiales de Alicante, Universidad de Alicante, Apartado 99, E-03080 Alicante, Spain; 2Instituto de Investigación Sanitaria y Biomédica de Alicante (ISABIAL), E-03080 Alicante, Spain

**Keywords:** SiC, coke, industrial synthesis

## Abstract

SiC is one of the most important ceramics at present due to its excellent properties and wide range of applications. The industrial production method, known as the Acheson method, has not changed in 125 years. Because the synthesis method in the laboratory is completely different, laboratory optimisation may not be extrapolated to the industrial level. In the present study, the results at the industrial level and at the laboratory level of the synthesis of SiC are compared. These results show that it is necessary to make a more detailed analysis of the coke than the traditional one; therefore, the Optical Texture Index (OTI) should be included, as well as the analysis of the metals that form the ashes. It has been found that the main influencing factors are OTI and the presence of Fe and Ni in the ashes. It has been determined that the higher the OTI, as well as the Fe and Ni content, the better the results obtained. Therefore, the use of regular coke is recommended in the industrial synthesis of SiC.

## 1. Introduction

Historically, petroleum coke production has been concentrated into two main areas: the manufacture of carbon anodes in the aluminium smelting cells and the production of electrographites for the steel industry [1]. Nowadays, the reduction in the number of smelting cells and the crisis in the steel industry have led to a steady decline in the consumption of coke. It is therefore necessary to develop new markets for petroleum coke, particularly in the field of non-oxidic ceramics (i.e., SiC). Silicon carbide is commercially produced by the reaction of high-grade silica sand or quartz with carbon in an electric resistance furnace (Acheson method) [2]. The most commonly used carbon precursor is green coke. Two different types of silicon carbide are produced by this method, depending on the purity of the raw material: black SiC (low purity) and green SiC (high purity). 

Normally, in the SiC industry, the coke requirements are not very restrictive: only a maximum in ash and sulphur content and a minimum in the fixed carbon content. The aim of the present work is to show that it is necessary to know more about coke properties in order to improve the quality and productivity of SiC production.

## 2. Theoretical Background 

### 2.1. Industrial Production of SiC

SiC is still produced in a similar way to the Acheson original 1897 patent [2]. This is based on passing an electric current through a bed of sand with a carbon electrode, where he found that a new, extremely hard material adhered to the electrode, which he called carborundum, the name of which has survived to this day, despite the fact that what he thought he obtained was a combination of carbon and aluminium (carbon + alundum). The other proper name that it receives is moissanite, since the French geologist Henry Moissan [3] discovered it while studying certain meteorites.

The process is very simple: heat a mixture of sand (SiO_2_) and green coke (C). The mixture is heated to a temperature above 1900 °C, and the following reactions take place:SiO_2_ + C → SiO + CO; SiO + 2C → SiC + CO

These reactions may also occur:SiO_2_ + CO → SiO + CO_2_; CO_2_ + C → 2CO

For many years, NaCl and sawdust were added to purify the sample by forming volatile chlorides such as Fe and Ti with the typical impurities of sand and sawdust to create spongier mixtures and facilitate the escape of gases. It is not currently a widespread practice due to the high environmental impact and corrosion of equipment.

Figure 1 shows a diagram of the synthesis process, where we can highlight the furnace. The furnace is an open furnace, usually with a capacity of about 100 tonnes, with a graphite electrode in the centre and a mixture of sand and coke around it. An electric current is simply passed through the electrode, which, due to the Joule effect, heats up to temperatures of 2500 °C for 24–48 h, creating a radial temperature gradient, so that it does not react or only partially reacts in the outermost part of the bed. In this process, we can highlight two things: one is that it is a discontinuous process, and the other is that energy consumption is very high, for which the companies have installed primary sources of energy readily available from the first factory (Carborundum, Niagara Falls [4]). Figure 1 also shows green SiC and black SiC, the main difference being the impurity content; the standard purity of black SiC is 98% (the main impurity is Al), while that of green SiC is over 99%. It is usually assumed that there is only one phase, but in reality, there are many. In a titanic work by Knimberg [5], he discovered that there are more than 170 polytypes, although the most abundant is 6H (Ramsdell notation [6]).

### 2.2. Laboratory Production of SiC

The production in the laboratory is completely different; in this case, the mixture of the precursors is made, placed in an alumina boat, and heated between 1400 and 1600 °C in an inert atmosphere, usually Ar [7,8,9,10,11,12,13], for periods ranging from a few minutes to a few hours. In this case, we obtain beta-SiC, which is now a single polytype, 3C. Because the manufacturing process is completely different (in terms of temperatures, times, and working atmosphere), it is not obvious that the results obtained in the laboratory are 100% transferable to the industrial level.

The excellent properties of SiC (low density, high decomposition point, and good mechanical properties up to 1600 °C) have made it a ceramic with many applications [14,15,16], and it is the most widely used for the reinforcement of composite materials [17,18,19,20]. It is also fundamental in the electronics industry, and today we have it in a multitude of applications, where we can highlight its use in Stocky diodes that are installed in most current cars [21,22,23,24], and its use has even been postulated for its use in 2D electronics, in the graphene/SiC system [25,26,27]; it remains the reference material for use of nuclear energy generators, especially in fusion reactors [28,29,30]. For this reason, research has been very intense, not only at the level of powder production but also at the level of whiskers [31,32,33,34] and fibres [35,36,37,38,39] or in the search for new precursors [40,41,42,43,44,45].

The group of Prof. Narciso has studied the effect of the type of coke in the production of SiC, first in the reaction with silica [46] and later in the direct reaction with silicon [47,48]. In both cases, it is evident that the higher the optical texture index (order), the higher the reactivity of the carbon to form SiC. Similar results have been obtained by Professor Eustathopoulos, who analyses from HOPG (Highly Oriented Pyrolitic Graphite) to GC (Glassy Carbon) [49,50,51]. It is therefore evident that, at least at the laboratory level, there is a clear effect between the structural order in the coke and the reactivity in the formation of SiC.

### 2.3. About the Coke

Coke is the solid residue obtained from the pyrolysis process of petroleum residues (i.e., at 500 °C and 1 MPa). The microstructure of the coke is defined during the carbonisation process (distillation of the petroleum feedstock and mesophase formation-coalescence-orientation), and it has been studied by polarised reflected light microscopy [46,52]. Very reactive feedstocks and/or drastic conditions produce a very small optical texture (mosaic texture). Aromatic feedstocks and/or soft carbonisation conditions produce coke with large flow domains and intermediate optical texture domains (see Figure 2) [46,53]. 

It is recognised that carbons and graphite differ in their reactivity according to their structural order (optical texture); the more highly ordered graphite is generally less reactive than the disordered crystalline carbons [54,55]. At the microscopic level, reactivity in air is a function of the quality of the coke [56,57]. Thus, green coke with an optical texture composed of mosaics and small domains will have a disordered microstructure, and the entire surface will react with air. If the structure is well ordered (flowing domains), gasification will take place at imperfections such as cracks.

Usually coke is classified into four major groups: (i) combustion, mosaic texture, and a sulphur content near 6% and used for energy production, (ii) regular, medium domain texture, and a sulphur content between 1 and 2%, used in the production of aluminium (anodes of smelting cells), (iii) needle, only in the flow domain and needle superpremium (top quality) in the very large flow domain, with a sulphur content below 0.5%, used in the production of electrographites, and (iv) shot, where texture is not important only the total content of impurities is below 0.5%, and is used for increasing the carbon quantity in the steel.

## 3. Materials and Methods

The synthesis in the laboratory followed the classical processes [53]. The SiO_2_ is mixed with the carbon material in a 60/40 weight ratio (SiO_2_/C-_fixed_), and the mixture is placed in an alumina boat and heated at 1500 °C for 3 hours in a dynamic Ar atmosphere (100 mL/ min). To determine the degree of advancement of the reaction, the obtained material is treated at 800 °C in a muffle furnace to remove unreacted carbon, and the residue is treated with a mixture of concentrated acids in a ratio of 10/1 (HF/HNO_3_) to remove unreacted silica, and possible impurities. The industrial synthesis was carried out according to the procedure described in the introduction, without the use of sawdust or NaCl. The furnace used has a capacity of 25 tonnes.

The materials used are silica and green coke, whose characterisation are summarised in Table 1 and Table 2.

Characterisation

Crystallographic phases were identified by powder X-ray diffraction (PXRD, Austria) recorded on a Brucker D8-Advanced diffractometer (Billerica, MA, USA) with a Goebel mirror and a Kristalloflex K 760-80F X-ray generation system, fitted with a Cu cathode and a Ni filter. Spectra were registered between 5° and 80° with a step of 0.05° and a time-step of 3 s. The characterisation of the samples by means of a TG-DTA-MS was carried out on a TGA/SDTA851e/LF/1600 from Mettler Toledo (Columbus, OH, USA) equipped with the Thermostar GSD301T Pfeiffer mass spectrometer. The TG experiments were carried out in a dynamic atmosphere of Ar (100 cm^3^/min) and/or in a dynamic O_2_/Ar (1/4) atmosphere with a heating rate of 10 °C/min while scanning masses up to 200 amu. A Thermo Scientific CHSN TM FlashSmart Elemental Analyser (Waltham, MA, USA) was used to determine the fixed carbon and sulphur content, and for the determination of volatile material, we followed the procedure proposed by Linares et al. [57]. X-ray fluorescence was used for mineralogical analysis of the coke residues after calcination at 850 °C for 12 h. A PHILIPS MAGIX PRO (PW2400) (Amsterdam, The Netherlands) was equipped with a rhodium tube and a beryllium window. Ground coke samples were mounted in a cold-setting epoxy resin, polished following the standard techniques, and examined using a polarised-light microscope with a half-wave retarded plate (Leica DM1750 M, Wetzlar, Germany). Quantitative assessments of the optical texture (OI) of the coke were assessed using a point-counting technique based on 500 points (see [45] and the Appendix A for a detailed procedure). To determine the IR spectra of the cokes, a BRUKER IFS 66/S spectrometer was used, capable of working with a resolution of up to 1 cm^-1^. The detector used was a DLaTGS at room temperature.

## 4. Results and Discussion

Using XRD, it was observed that there were no differences in the diffraction patterns of all the performed syntheses. In all cases, the beta-SiC for the synthesis in the laboratory was obtained, and the alpha-SiC for the industrial synthesis was as would be expected. In Figure 3, we observed five peaks associated with the 3C phase; in addition to the peak corresponding to the plane (1,1,1), we observed a shoulder, which is associated with defects in SiC growth.

The IR spectra of the different cokes are very similar, with only the intensity of two significant bands changing. Figure 4 shows the typical spectra of the coke materials. The most important bands are: 3050 cm^−1^ stretching vibrations of the C–H bond in the aromatic ring, 2960 cm^−1^ asymmetric stretching vibrations of the C–H bond in the methyl groups, 2920 cm^−1^ asymmetric stretching vibrations of the C–H bond in the methylene groups, 2860 cm^−1^ symmetric stretching vibrations of the C–H bond in the methyl and methylene groups, and 2720 cm^−1^ stretching vibrations of the C–H bond in the bridge structures (CHAr_3_).

The intensity ratio between the band at 3050 cm^−1^ and the bands in the interval of 2800–3000 cm^−1^ is representative of the ratio of aromatic/aliphatic groups, which is directly related to the reactivity of coke. Reactivity increases with the number of aliphatic groups [54]. Table 3 shows the ratio of aromatic/aliphatic groups. It is observed that the increase in aromatic content increases the performance of the formation of SiC following the laboratory synthesis. This is not a direct relationship; it is only a trend, as the formation of SiC does not only depend on the structure of coke. Impurities play a key role, especially Fe, Ni, Co, and V [55]. 

Typical TG-DTA profiles are shown in Figure 5. The DTA shows an asymmetric peak in both cokes. Normally, the asymmetry is derived from the two combustion processes [58]. The TG profile also shows two zones. Probably the first zone is related to the most reactive coke as a mosaic and the second to the less reactive coke as domains and/or flow domains. Two distinct peaks were not observed in any of the samples. Either it is observed as in Figure 4 or only a single peak appears. We have assumed that this asymmetry (shoulder) is due to the presence of two types of carbon, since in either case the structure is perfectly homogeneous. Table 3 shows the temperature of the larger peak. A good relationship between peak temperature and yield can be observed in the laboratory samples as well as the industrial samples. This relationship is not perfect. It is like a trend, similar to IR data, but it is very simple and quick to make. This relationship is very important because no one has reported anything similar. It is important to note that all the cokes used in the industrial test complied with the parameters required for the production of SiC (sulphur content, volatile matter, fixed carbon, and ash).

Narciso et al. proposed a multivariate analysis to determine the progress of the SiC formation reaction [46]. From an exhaustive analysis of all the variables that could influence it, we determined that the only ones that are relevant are the OTI, the content in (Fe + Ni), and the content in V. The presence of alkali, alkaline earth, or even Ti does not affect the performance. In this research, the same parameters have been used directly with a slight difference since it has been shown that Ni is more active than Fe in the synthesis of SiC [32].

The proposed equation is as follows:Yield (%) = a.(OTI^1/2^) +b.(%Ni) +c.(%Fe) +d.(%V)
where a, b, c, and d are the tunable parameters, which are slightly different from those originally proposed:Yield (%) = a.(OTI^1/2^) +b.(%Ni+%Fe) +c.(%V)

After multivariate adjustment, we find that the parameters are: a = 8, b = 36, c = 34, and d = 3. The parameters are practically the same as those originally proposed (a = 7.8, b = 34.5, and c = 3.1), which shows the validity of this proposal.

Figure 6 shows the relationship between the experimental data and the data obtained from the multivariate analysis. It can be seen that the slope is one and that all the data fit perfectly, which gives us an idea of the efficiency of the proposed equation. The analysis of the data obtained at the industrial level is slightly more complex. In this case, it is proposed to assign a numerical value to the adjective (bad = 1, medium = 2, medium good = 3, good = 4, and very good = 5). Figure 7 shows the relationship between the "industrial yield" and the yield in the laboratory. Obviously, we cannot obtain a perfect relationship, but we can observe a very clear trend, which once again highlights the need for a more thorough analysis of coke to decide whether it is optimal for SiC production.

It seems obvious that it would be convenient to use regular coke for a number of reasons. The main one is that the two cokes whose microstructure and composition are close to regular coke have given good results at an industrial level. The V content is very low, so it would avoid possible contamination in the manufacture of certain ceramic materials, as they would leave orange stains in the ceramic materials, so it is imperative, at least for green SiC, to avoid cokes from high-content oils such as that from Venezuela. Additionally, from an environmental point of view, the emission of S compounds into the atmosphere is reduced by 85%. Already 60 years ago, Clayton [59] made a similar recommendation, where he showed that by using graphite instead of coke in the synthesis using the Acheson method, it improved the performance by 50%, although this study was performed in the laboratory. At that time, it was something radical, since the price of electricity was low, especially if the SiC was produced in conjunction with a primary source of energy. Additionally, the price of graphite was very high. However, it is important to point out that the way to improve productivity is to use more graphitisable coke, as this research has shown.

## 5. Conclusions

The following conclusions emerge from the present investigation. The traditional criteria for the selection of cokes for the production of SiC are not sufficient, as shown by the industrial results. The structure of the coke is the most relevant parameter; the more ordered the coke, the higher the yield obtained. The presence of metals such as Fe and Ni catalyses the formation reaction of SiC, especially at the laboratory level; other metals such as alkaline or alkaline earth metals do not affect it, and V has a lower catalytic effect. Proposing the following equation: yield (%) = 8.(OTI^1/2^) + 36.(%Ni) + 34.(%Fe) + 3.(%V)

It is proposed that the most appropriate coke is regular coke since the results are optimal. An additional advantage is related to the S content; it is much lower, therefore it generates fewer environmental problems.

It is convenient to carry out a previous study of the coke in greater depth, analysing the metals it contains and the optical texture index.

## Figures and Tables

**Figure 1 materials-16-02034-f001:**
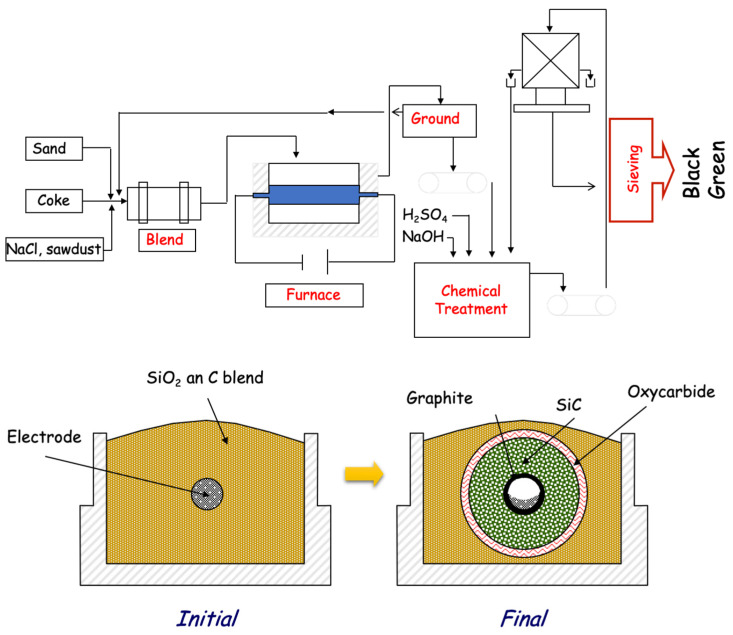
(**top**) Diagram of the Acheson process for obtaining SiC. (**bottom**) A detail of the furnace displaying the radial distribution of the products.

**Figure 2 materials-16-02034-f002:**
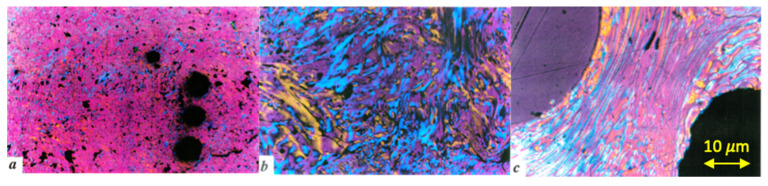
Optical photomicrographs of coke structures. (**a**) Mosaic, (**b**) domains, and (**c**) flow domains.

**Figure 3 materials-16-02034-f003:**
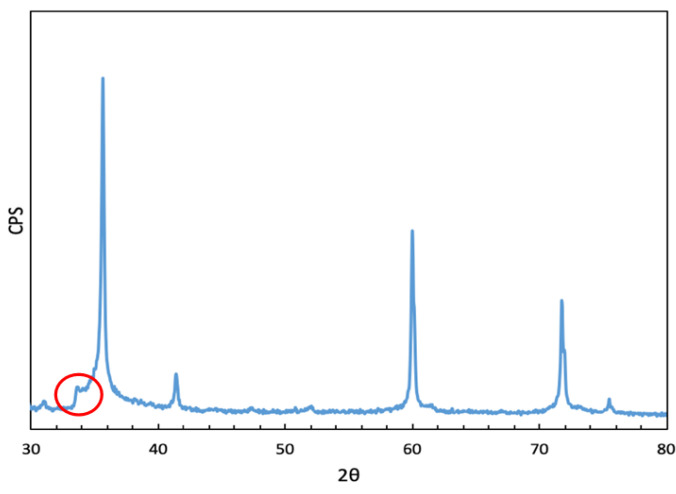
Diffractogram pattern of beta-SiC. The circle indicates a shoulder that is associated with defects, which appears as a shoulder at the peak (1,1,1).

**Figure 4 materials-16-02034-f004:**
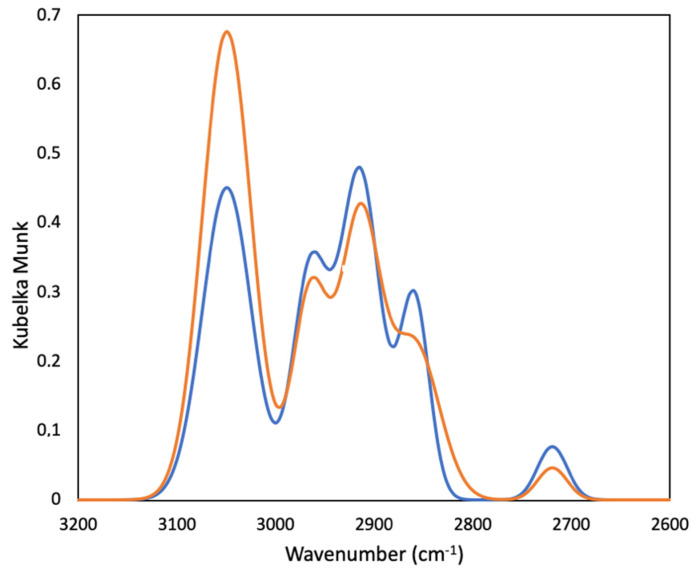
FTIR spectra of coke number 1 (red) and number 2 (blue).

**Figure 5 materials-16-02034-f005:**
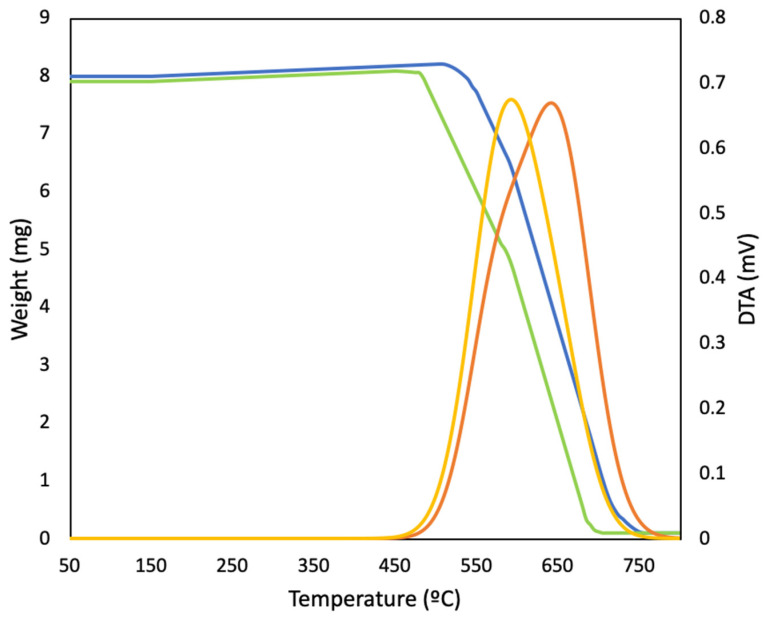
TG-DTA profiles of coke number 1 (red-blue) and coke number 2 (green-yellow).

**Figure 6 materials-16-02034-f006:**
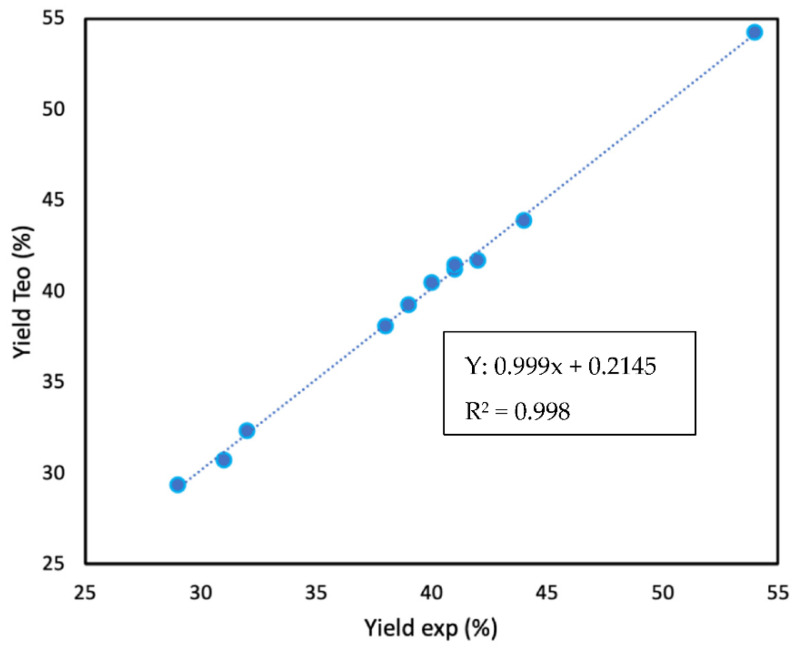
Experimental vs. theoretical yield. The data for the theoretical SiC yield were estimated from the following equation: yield (%) = 8.(OTI^1/2^) +36.(%Ni) +34.(%Fe) +3.(%V).

**Figure 7 materials-16-02034-f007:**
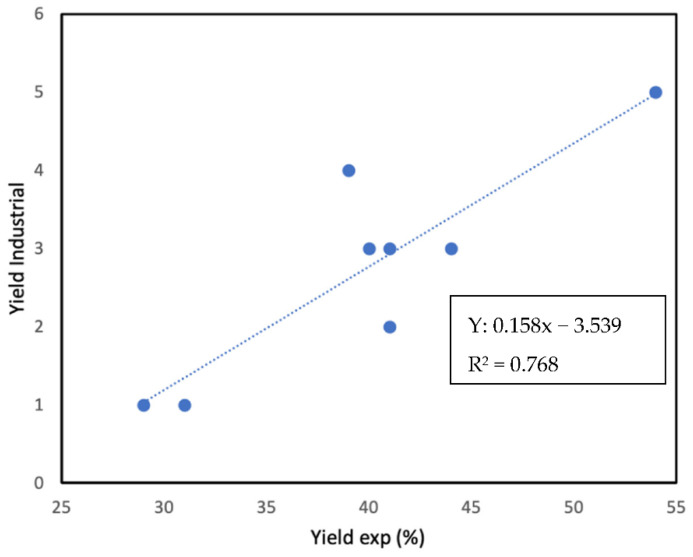
Experimental vs. industrial yield.

**Table 1 materials-16-02034-t001:** Silica analysis (wt%).

SiO_2_	Fe_2_O_3_	TiO_2_	Al_2_O_3_
99.8	0.025	0.020	0.060

**Table 2 materials-16-02034-t002:** Analysis of different green petroleum cokes used in this study.

*Coke*	*Volatile (wt%)*	*Ash (wt%)*	*Fixed C (wt%)*	*Sulphur (wt%)*
*1*	9.76	1.14	89.10	6.21
*2*	7.22	0.78	92.10	5.31
*3*	10.11	0.44	89.45	5.70
*4 ^@^*	6.64	0.12	93.24	1.40
*5*	8.00	0.44	91.56	6.38
*6 ^@^*	8.52	0.21	91.27	1.54
*7*	9.43	0.62	89.25	3.83
*8*	9.53	0.37	90.10	4.03
*9*	8.90	0.45	90.65	4.85
*Regular*	9.95	0.15	93.76	1.09
*Combustion*	12.34	0.30	87.44	6.05
* **Coke** *	**Fe_2_O_3_ (wt%)**	**NiO (wt%)**	**V_2_O_5_ (wt%)**	**CaO (wt%)**
*1*	0.250	0.291	0.142	0.001
*2*	0.011	0.010	0.200	0.000
*3*	0.011	0.019	0.201	0.001
*4 ^@^*	0.021	0.029	0.000	0.000
*5*	8.00	0.44	91.56	0.002
*6 ^@^*	0.049	0.098	0.051	0.001
*7*	0.102	0.205	0.109	0.003
*8*	0.084	0.150	0.150	0.002
*9*	0.151	0.199	0.102	0.001
*Regular*	0.03	0.004	0.02	0.001
*Combustion*	0.06	0.06	0.145	0.000

^@^ Used to manufacture green SiC.

**Table 3 materials-16-02034-t003:** Proposed additional characterisation of coke used for the manufacture of SiC and yield at both laboratory and industrial levels. OTI is the Optical Texture Index [46].

Coke	OTI	Ratio aro/ali	DTA peak ( °C )	Yield (%)	Industrial
*1*	19	1.3	650	54	Very Good
*2*	13	0.9	600	31	Bad
*3*	12	0.9	584	29	Bad
*4^@^*	22	1.4	689	39	Good
*5*	19	1.2	665	40	Med–Good
*6^@^*	20	1.3	698	41	Med–Good
*7*	17	1.2	655	44	Med–Good
*8*	13	1.0	603	38	Not used
*9*	13	0.9	595	41	Medium
*Regular*	24	1.2	702	42	Not used
*Combustion*	12	0.9	590	32	Not used

## Data Availability

Not applicable.

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
