# Peer review of "Manufacture of SiC: Effect of Carbon Precursor"

_materials, 2023, doi:10.3390/ma16052034_

Round 1

Reviewer 1 Report

This paper has illustrated the effect of carbon precursor on SiC manufacture, which is important and helpful. However, there are several questions should be clarified before acception.

1.       Will the carbon precursor affect the element ratio of SiC? If yes, will it be Si rich or C rich?

2.       Carbon residue at the interface between SiC and dielectrics is a serious problem in SiC device fabrication, and several methods such as plasma modification (Applied Physics A, 2022, 128:941) and annealing (Applied Physics A, 2022, 128:1132) have been proposed to suppress it. Will the carbon precursor on SiC manufacture aggravate this problem? I suggest the authors to add some comments and references to clarify this point.

Author Response

I have attached a l

Dear reviewers,

First of all, thank you for taking the time to review this manuscript. We have carefully read your comments and have taken them into account when modifying the manuscript.

We would like to point out that this is the first time that industrial data regarding SiC production is included in a scientific article. The industrial experiments are with 25,000 kg as indicated in the text. The data has been collected over several years. And at an industrial level, what we have is whether it is good, bad, very good……., which is very difficult to capture in a scientific article.

When we recommend regular coke as coke, we base ourselves on the fact that it is a coke with great production, that the OTI is high, the price is not excessive, and the sulfur content is of the order of 1%.

I hope this new manuscript is suitable for you.

Reviewer 1.

Will the carbon precursor affect the element ratio SiC? If yes, will it be Si rich or C rich?

We have not found any difference with the type of carbon we are using.

Here we are talking about the SiC that is obtained is for use in the abrasives and composite materials industry, and each batch is 300,000 Kg. When we talk about the electronics industry, the growth procedures are different, and the cost of SiC is almost 3 to 4 orders of magnitude higher.

Added a comment in the introduction reference 24.

Reviewer 2 Report

In this manuscript, the authors present the relationship between SiC synthesis yield and properties of source coke. They found that coke’s Optical Texture Index and contamination level (Ni, Fe, V) contribute to the yield. The conclusion may help manufacturers select the best coke for their SiC production by the Acheson process. However, the authors should do a revising after addressing the following questions.

  1. Please describe the OTI determination in your manuscript.
  2. Optical images for each kind of coke in your manuscript are recommended.
  3. FTIR spectra and TG-DTA profile in Fig.4 and Fig.5 are of coke No2 and No1, respectively. What about all other kinds of coke?
  4. The axis labels in Fig. 6 and Fig. 7 confused me. What are “Yield Teo” and “Yield exp”? Are they “Regressed Yield” and “Laboratory Yield”, respectively?
  5. How did the equations in lines 250 and 253 proposed and obtained?
  6. Why was CaO ignored?
  7. Some kinds of coke have a high yield in the laboratory but a low yield in industrial production, (e.g., No9), could you explain it in more detail?
  8. Please revise English typos.

Author Response

Dear reviewers,

First of all, thank you for taking the time to review this manuscript. We have carefully read your comments and have taken them into account when modifying the manuscript.

We would like to point out that this is the first time that industrial data regarding SiC production is included in a scientific article. The industrial experiments are with 25,000 kg as indicated in the text. The data has been collected over several years. And at an industrial level, what we have is whether it is good, bad, very good……., which is very difficult to capture in a scientific article.

When we recommend regular coke as coke, we base ourselves on the fact that it is a coke with great production, that the OTI is high, the price is not excessive, and the sulfur content is of the order of 1%.

I hope this new manuscript is suitable for you.

Reviewer 2

  1. Please describe the OTI determination in your manuscript. Done

2.Optical images for each kind of coke in your manuscript are recommended. I have not included them because it does not contribute anything, and I have include 3 extreme photos so that the reader knows what we are talking about. In a coke we almost always have all the structures, the difference is the percentage we have of each of them.

3.FTIR spectra and TG-DTA profile in Fig 4, and FiG. 5 are coke N02 and No 1, respectively. What about all other kinds of coke. We think that including all the cokes in a graph does not contribute anything, that what is important is the data that can be extracted from it. We have modified the graphs and we have included two so the reader can compare a little more directly.

4.The axis label in Fig 6 and Fig 7 confused me. What are “yield Teo” and “Yield exp”? Are they Regresses Yield and Laboratory Yield respectively? Yes. In a multivariate analysis, it is the easiest method to check the goodness of the analysis. If it were perfect, the line would be Y=x and the coefficient R2=1. It is observed that there are no points outside the model and that it is practically Y=x. Figure 7 is different, since we have assigned values to the industrialist to be able to compare with the experimental result, this graph is a visual guide, where we can only observe trends

  1. How did the equation in lines 250 and 253 proposed are obtained? The SIMPLEX method of multivariable analysis has been used, as indicated in the reference 46. In said reference, a much more complex analysis is made to determine which variables have more weight, and in this manuscript it is taken as the equation of starting that equation.
  2. Why was CaO ignored? It has not been ignored, simply that it is not relevant, in the reference 46 it was already revealed that it had no weight, and there were cokes with a significant alkaline and alkaline earth content. In this case, with such a low content, it does not affect anything
  3. Some kinds of coke have a high yield in the laboratory but low yield in industrial production (e.g. No)) could you explain in more detail? You are right, there is an error in the table, it is not 90%, it is 41%.
  4. Please check English typos. done

Reviewer 3 Report

The manuscript entiteled ‘Manufacture of SiC: Effect of Carbon Precursor’ compared the results at the industrial level and at the laboratory level of the synthesis of SiC. It has been obtained that the higher the OTI and the Fe and Ni content, the better results are obtained. Therefore, the use of regular coke is recommended in the industrial synthesis of SiC. The authors should address and implement the following comments before the manuscript may be accepted for publication.

1.In figure 7, R2=0.7682. Please illustrate the meaning of the relationship between ‘industrial yield’ and yield in the laboratory.

2.Some quantitative results may be added to the conclusion section.

Author Response

Dear reviewers,

First of all, thank you for taking the time to review this manuscript. We have carefully read your comments and have taken them into account when modifying the manuscript.

We would like to point out that this is the first time that industrial data regarding SiC production is included in a scientific article. The industrial experiments are with 25,000 kg as indicated in the text. The data has been collected over several years. And at an industrial level, what we have is whether it is good, bad, very good……., which is very difficult to capture in a scientific article.

When we recommend regular coke as coke, we base ourselves on the fact that it is a coke with great production, that the OTI is high, the price is not excessive, and the sulfur content is of the order of 1%.

I hope this new manuscript is suitable for you.

Review 3

  1. In figure 7 R2=0.7682. Please illustrate the meaning of the relationship between “industrial yield and the yield in the laboratory. The idea is to show in a graphic way if what we do in the laboratory is relevant to what is observed at an industrial level. Keep in mind that I have given some numerical values to industrial appraisals that are adjectives, "very good", "bad"...
  2. Some quantitative results may be added to the conclusion section. done

Round 2

Reviewer 1 Report

I suggest to accept this manuscript.

Author Response

thank you very much

Reviewer 2 Report

Thank you for your revision.

I think you could improve some minor things as follows:

1. Fig. 3 needs more denotations, explanation, and discussion, e.g. peak locations, plane, etc. In the current form, the Fig does not give much information.

2. Similarly Fig 4 & 5

3. Table 3 and Fig.7 need qualitative values for industrial yields.

Author Response

Comments are attached to a file.
